# A Parking Space Allocation Method to Make a Shared Parking Strategy for Appertaining Parking Lots of Public Buildings

**Yifei Cai** [1] , **Jun Chen** [1,*], **Chu Zhang** [1] **and Bin Wang** [2]

1 School of Transportation, Southeast University, Nanjing 210096, China; 13675102961@163.com (Y.C.); zhangchu0720@gmail.com (C.Z.)
2 Urban and Traffic Planning Design Institute, Shanghai Urban Construction Design and Research Institute Co., Ltd., Shanghai 200125, China; wangbin@sucdri.com
* Correspondence: chenjun@seu.edu.cn

**Abstract:** Appertaining parking lots of public buildings provide a large proportion of parking supply in cities. However, these parking lots mainly serve the parking demands of public buildings, leading to a low utilization ratio of parking spaces. It is therefore required to implement a shared parking strategy for these parking lots. In this study, a parking space allocation method (PSAM) at the network level is proposed to allocate the parking demand to a parking lot and then the parking space. The users are divided into M-users (users of the buildings) and P-users (public users). The shared parking strategy is analyzed from the aspects of open window, parking fee, and ratio of reservation spaces. The users are allocated to a parking lot by a multinomial logit(MNL) model. Specifically, it is determined whether they can enter parking lot and which space they are allocated according to the specific rules. After all the users are allocated with a parking space, the rejection number of M-users, occupancy rate, and profits of each parking lot are collected and a NSGA-II (non-dominated sorting genetic algorithm II) algorithm is designed to determine the optimal strategy for each parking lot according to the above. Compared with the results of all-time all-space shared parking strategy, our method shows better performance in balancing the interests of all appertaining parking lots and protecting the interests of M-users while obtaining considerable profits for the parking lots.

**Keywords:** Shared parking; appertaining parking lot; parking space allocation; parking fee; parking space reservation

## 1. Introduction

At present, the rapid increase in car ownership in China has far exceeded the supply capacity of urban parking lots [1,2]. Forexample, in 2017, the total number of parking spaces in Beijing was 3.82 million, with 1.95 million residential parking spaces, 1.47 million appertaining parking spaces, and 0.4 million other parking spaces. The shortage in the parking supply was 1.91 million. Meanwhile, the average utilization ratio of parking space was lower than 50%,withthe utilization ratio of appertaining parking lots being less than 40% at night [3].

Faced with the problems of parking supply shortage and low utilization ratio, it is required to share the vacant time of parking spaces with the users in need [4–7]. The idea of shared parking was very popular when it was newlyproposed, and shared parking strategies started to emerge in residential communities [8–10]. The basic idea of the strategies is that a parking space owner sells the vacant time of theirspace directly or via an intermediary agent to public users on an e-parking platform. However, the effects of the shared parking strategies seem unsatisfactory, and they mainly have the following problems:

(1)   The parking supply in residential communities is insufficient when the residents with parking space still rent out their space to public users, probably causing dissatisfaction among residents who do not have parking spaces.

(2)   The entry of public users causes chaos in the parking management and the income is too low for the residents totake the risk to rent out their parking spaces.

(3)   As the parking supply in residential communities is concentrated in the daytime and scattered in different communities, itisdifficultforpublic users to find a space they need.

Shared parking is a good way to solve the parking problem, however it is obvious that the scattered shared parking spaces in residential communities cannot provide enough parking spaces at any time [11]. However, the appertaining parking lots of public buildings can solve this problem. Different from residential communities, the parking spaces in appertaining parking lots are managed by one parking manager. Since the manager cannot realize the impact of implementing shared parking on the M-users(users of the buildings), it is hard to promote shared parking in these parking lotsin an enterprise-led way.

For appertaining parking lots, the government-led promotion which aims at maximizing the interests of all parking lots in a specific area is clearly a better approach. If each appertaining parking lot in a certain area iswithin an acceptable walkingrange, it is possible for a user to park in any parking lot in this area. Since the parking demand differs in each parking lot, the peak and valley time of the utilization ratio in each parking lot may complement or compete with each other. When one parking lot changes its parking fee or other variables, it may influence the demands of other parking lots. If one parking lot is too occupied to provide a space for the M-users, although other parking lots in this area provide more benefits, for this parking lot, the shared parking is unacceptable; if this parking lot quits implementing shared parking, the parking distribution in other parking lots will change as well. Therefore, to persuade enough appertaining parking lots to implement shared parking, it is necessary to consider the problem of parking space allocation at the network level and show them the benefits that are brought by implementing a shared parking strategy.

Before implementing shared parking, these parking lots mainly satisfy the demands (M-user) of the buildings they are appertained to. For government parking lots, most users are government employees. Public users (P-users) are not allowed to enter the government parking lots. For hospital parking lots, although spaces are provided for P-users, the parking fee is so high that P-users would rather find an on-street parking space [12]. Therefore, to implement shared parking in these parking lots, it is first necessary to satisfy the demands of M-users, then P-users.

A novel parking space allocation method is proposed in this research. In this method, the open window is determined at the valley time of the parking lot before shared parking [13] and the parking fee [14–18] in each open window is considered as the variable to control the P-user's demand. To prevent too many P-users from parking in a parking lot, enough spaces should be reserved for M-users [19–21]. Before all the non-reserved spaces are occupied, no users are allowed to park in reserved spaces. Besides, only M-users are allowed to park in reserved spaces after all the non-reserved spaces are occupied. When all the parking lots implement their parking strategy, the users will be allocated to the target parking lot based on their personal attributes, travel time, and parking fee. When the user arrives at the target parking lot, it will be determined whether the user can enter and which space will be allocated based on user type (M-user or P-user) and the occupancy situation of the parking lot (whether non-reserved space and reserved space are occupied). If rejected, the user has to find another space in other parking lots. When all the users are allocated to a space, the rejection number of M-users, parking profits, and occupancy rate of each parking lot will be collected. An NSGA-II algorithm is designed to calculate the optimal strategy for each parking lot according to these indices.

The rest of this paper is structured as follows. In Section 2, the parking space allocation method is illustrated, with the function of allocating one user from origin to its final parking space. Based on this method, an NSGA-II algorithm is designed in Section 3 to determine the optimal strategy for

each parking lot. In Section 4, a parking lot network is established and the results of our method are compared with those of the traditional strategy to show the advantage of the proposed method.

## 2. Parking Space Allocation Method(PSAM)

This study involves M-users, P-users, reserved parking spaces, and non-reserved parking spaces. It is necessary to find a way to allocate users to a parking space rather than a parking lot. Therefore, a method which considers both the user attributes and parking data is proposed, with the function of allocating a user to a specific parking space of a parking lot. This is called the parking space allocation method (PSAM)in this paper.

The PSAM consists of four steps:

1.  Before implementing shared parking, the open window of each parking lot is determined using the gate data.
2.  Suppose all the parking lots start a shared parking strategy, they will need to make a set of decisions based onthe open window, the parking fee in each open window, and the reservation ratio. After that, the new parking demand of each origin is predicted.
3.  Allocate the parking demand to the parking lot from the origins based on the M-users and P-users. Determine whether they can enter the lot and which space they will be allocated to. If they cannot enter, then these users will be allocated to other parking lots until they find a parking space.
4.  After all users are allocated to a parking space, collect all the related indices of each parking lot, and use the NSGA-II algorithm to determine the optimal shared parking strategy.

### 2.1. Parking Resource Matrix (PRM) Method

The data used in this study is gate data from appertaining parking lots. The gate data include the arrival time *at* and the leave time *lt*. One gate datum corresponds to one user. As the gate data cannot show the specific parking space in which the user chooses to park, it is possible to number the parking space and allocate the gate data into a vacant parking space in the sequence of space number. A parking space is occupied in the period from time *at* to time *lt*. Under this assumption, a parking resource matrix (PRM) is proposed.

A PRM describes the occupancy situation of each parking space of a parking lot in the whole time period. The PRM of parking lot $n$ is denoted as $PRM_n$, which is a $p_n$ by $T$ matrix, where $p_n$ represents the number of parking spaces in the parking lot $n$ and $T$ is the total time period.

$$PRM_n = \begin{bmatrix} a_{11} & a_{12} & \ldots & a_{1T} \\ a_{21} & a_{22} & \ldots & a_{2T} \\ \ldots & \ldots & \ldots & \ldots \\ a_{p_n1} & a_{p_n2} & \ldots & a_{p_nT} \end{bmatrix} \tag{1}$$

where $a_{ij}$ is the occupancy situation of the $i^{th}$ parking space at time $j$. When $a_{ij} = 0$, it means the $i^{th}$ space is vacant at time $j$; when $a_{ij} = 1$, it is occupied.

Suppose one user $k$ needs to enter parking lot $n$, the arrival time is $at_k$, the parking time is $pt_k$, and the time step between $a_{ij}$ and $a_{i(j+1)}$ is $\Delta t$. Then, the number of columns occupied by parking time $pt_k$ in $PRM_n$ is given by:

$$gn= \text{round}(\frac{pt_k}{\Delta t}) \tag{2}$$

Suppose the $i^{th}$ parking space at time $at_k$ is vacant, which means $a_{i(at_k)} = 0$, then the result to allocate one user $k$ into $PRM_n$ is expressed as:

$$a_{i(at_k)}, a_{i(at_k+1)}, a_{i(at_k+2)}, \ldots, a_{i(at_k+gn-1)} = 1 \tag{3}$$

Equation (3) means the user $k$ occupies the $i^{th}$ parking space from time $at_k$ for a time period of $gn$.

The gate data set of parking lot $n$ is denoted as $pd_n(at_k, pt_k)$, $n = 1, 2, \ldots, N$, and $k = 1, 2, \ldots, K$. It covers the gate data of the whole time period $T$. To allocate all the gate data of parking lot $n$ to $PRM_n$, at time $t$, the data set is searched to find the user with the arrival time equal to $t$. Then, the $t^{\text{th}}$ column of the $PRM_n$ is searched in the sequence of parking space number. Two situations may happen: (1) A vacant space exists; (2) All spaces are occupied. In Situation 1, the user will be allocated to the first vacant space using the method above. Then go to allocate the next user. If all the users with arrival times equal to $t$ have been allocated, then $t = t + 1$. In Situation 2, it means this parking lot cannot accommodate more users, then $t = t+1$. The algorithm ends when $t > T$.

The specific algorithm is demonstrated in Algorithm 1:

---

**Algorithm 1**. The procedure of allocating gate data to the parking resource matrix (PRM).

---

**Input:** $pd_n, p_n, T$
**Output:** $PRM_n$
**Function** $PRM(pd_n, p_n, T)$
1. $PRM_n = zeros(p_n, T)$ % initialize the $PRM_n$ as a $p_n$ by $T$ zero matrix.
2. **for** $t = 1 : T$
3. **for** $k = 1 : K$
4. $i = 1$ % $i$ represents the parking space number.
5. **if** $pd_n(k, 1) = t$ % if the arrival time of user $k$ equals $t$.
6. $gn = round(\frac{pd_n(k,2)}{\Delta t})$; % calculate the number of columns $pt_k$ occupies
7. **end**
8. **while** $i <= p_n$ % search the space occupancy situation at time $t$ in the sequence of space number
9. **if** $PRM_n(i, t) = 0$　　　　% if $i^{\text{th}}$ space is vacant
10. $PRM_n(i, t : t + gn - 1) = 1$　　% allocate the gate data to the $i_{\text{th}}$ space, and occupy the space from $t$ to $t+gn$
$-1$
11. $i = i+1$ % with $i^{\text{th}}$ space is occupied, then determine whether $(i+1)^{\text{th}}$ space is occupied
12. **end**
13. **end**
14. **end**
**End Function**

---

This method of inputting all the parking data to $PRM_n$ is called the parking resource matrix method (PRMM) in this paper.

*2.2. Open Window*

After allocating all the gate data of parking lot $n$ to $PRM_n$, the number of occupied parking spaces at each time $t$ can be obtained and the valley time of the parking lot can be determined. Since the gate data is collected before implementing shared parking, the valley time of the parking lot can be viewed as the window time for other users after implementing shared parking. According to the study by Chen [22], the window time should meet two requirements:

1. The window time should be long enough; the minimum duration of window time is denoted as $ow_1$.
2. Enough parking spaces should be provided for P-users; the minimum number of spaces for P-users is denoted as $m_1$.

Therefore, after obtaining the number of occupied parking spaces at each time $t$, if there is a time period longer than $ow_1$ with the vacant spaces more than $m_1$, this time period is considered as one open window of the parking lot in this paper.

The following functions are defined:

accountH$_n(A(i, :))$: Count the number of all the columns with value $H$ in the $i^{\text{th}}$ row of matrix A.

accountH$_n(A(:, j))$: Count the number of all the rows with value $H$ in the $j^{\text{th}}$ columns of matrix A.

$con(A(i,:), ow_1)$: Find all the continuous numbers in the $i^{th}$ row of matrix A; if the length of the numerical string is larger than $ow_1$, then store the start and end numbers in a new matrix to act as the start time and end time of the open window.

The open window of parking lot $n$ is denoted as $OW_n$. The algorithm to determine the open window from an allocated $PRM_n$ is demonstrated in Algorithm 2:

---

**Algorithm 2**. The procedure of determining the open window.

---

**Input:** $PRM_n$
**Output:** $OW_n$
**Function** $OW(PRM_n)$
1. $i = 1$
2. **for** $t = 1{:}T$
3. $temp = account0_n(PRM_n(:,t))$ % count the number of vacant space at time $t$
4. **if** $temp >= m_1$ % if the vacant number is more than $m_1$ at time $t$
5. $A(1,i) = t$ % store time $t$ in the matrix A
6. **end**
7. **end**
8. $OW_n = con(A(1,:), ow_1)$ %find the time period which is longer than $ow_1$ and can provide more than $m_1$ spaces. The time period is viewed as the open window.
**End Function**

---

### 2.3. The Allocation of Parking Demand

Since the gate data cannot cover the new demands of the users after implementing shared parking, it is necessary to allocate the new users from the origins to the parking lot. After determining the optimal parking lot for these new users, the PRMM is used to allocate the new user to $PRM_n$. The allocation of the parking demand consists of two phases:

Phase 1: Allocate the users from origins to the prior parking lot with the highest probability using the logit model.

Phase 2: Use the PRMM to allocate the user to the $PRM_n$ of each parking lot. The M-users can park at any time in any space and the P-users can only park at the window time in non-reserved spaces. If a user cannot park in the parking lot, allocate it to other parking lots and repeat Phases 1 and 2 until a space is found.

#### 2.3.1. The Multinomial Logistic Regression Model

To allocate a user from its origin to a parking lot, a logit model is required in this paper. A multinomial logit (MNL) model is built according to the studies by [23,24] and the former work of our group [25]. The variables of the MNL model are divided into personal attributes and parking lot attributes, as shown in Table 1.

The calibration results of the MNL model are illustrated in Table 2.

The log likelihood function of the model, LL = −529.95969. The log likelihood function with only the constant term, LL(0) = −749.36712. The goodness of fit, $p^2$, is given by:

$$\rho^2 = 1 - \frac{LL(\beta) - K}{LL(0)} = 0.27 \tag{4}$$

The MNL model fits well, as shown by $p^2$ [26]. We can see that the choice of parking lot is negatively correlated with the parking fee, travel time, risk of no parking space, and perceived parking time. From the odds ratio, the parking fee shows the most significant influence on the users. When the parking fee increases by 1 level, the probability of a user choosing the parking lot will decrease by 54%.

**Table 1.** Variables of the multinomial logit model.

| Attributes | | Levels |
|---|---|---|
| Personal Attributes | Age (AGE) | 5 levels: ≤12 years old, 21~30 years old, 31~40 years old, 41~50 years old, and 51~60 years old. The value of each level is from 1 to 5, respectively. |
| | Driving experience (DE) | Five levels: ≤1 years, 2~3 years, 3~5 years, 5~10 years, and >10 years. The value of each level is from 1 to 5, respectively. |
| | Income (INC) | Four levels: ≤5000 RMB, 5000~10000 RMB, 10000~15000 RMB, and >15,000 RMB. The value of each level is from 1 to 4, respectively. |
| | Familiarity with the surrounding parking lot (FAM) | Three levels. The value of each level is from 1 to 3, respectively. |
| | Perceived risk of no parking space (RISK) | Five levels: ≤10%, 11~25%, 26~50%, 50~75%, 75~100% The value of each level is from 1 to 5 respectively |
| | Perceived Waiting time (WT) | Four levels: ≤2 min, 2~5 min, 6~10min, and >10 min. The value of each level is from 1 to 4, respectively. |
| Parking lot attributes | Parking fee (PF) | Four levels:4 RMB/h, 8 RMB/h, 12 RMB/h, and >12 RMB/h. The value of each level is from 1 to 4, respectively. |
| | Travel time (TT) | Four levels: ≤2 min, 2~5 min, 6~10min, and >10 min. The value of each level is from 1 to 4, respectively. |

**Table 2.** The calibration results of the multinomial logit model.

| Alternative-Specific Conditional Logit | | | | | Sample size = 558 | | | | |
|---|---|---|---|---|---|---|---|---|---|
| Case Variable: PID | | | | | Alts per case: | | | | |
| Alternative Variable: TYPE | | | | | min = 4; avg = 4.0; max = 4 | | | | |
| The Maximum Likelihood Function Value | | | | | Wald chi2(16) = 92.43 | | | | |
| Log likelihood = −529.95969 | | | | | Prob > chi2 = 0.0000. | | | | |

| | Variables | Coefficient | | Odds ratio | Standard deviation | Z value | P > \|z\| | [95% Conf. Interval] | |
|---|---|---|---|---|---|---|---|---|---|
| Parking lot attributes | PF | $\beta_1$ | −0.7705 | 0.4608 | 0.1836 | 4.20 | 0.000 | 0.4106 | 1.1303 |
| | TT | $\beta_2$ | −0.9756 | 0.6529 | 0.1462 | 6.67 | 0.000 | 0.6890 | 1.2622 |
| Personal Attributes | Age | $\alpha_1$ | −0.3695 | 0.7910 | 0.2055 | −1.80 | 0.072 | −0.7723 | 0.0333 |
| | DE | $\alpha_2$ | −0.2490 | 0.7795 | 0.1421 | −1.75 | 0.080 | −0.5276 | 0.02950 |
| | INC | $\alpha_3$ | 0.1293 | 1.1380 | 0.1744 | 0.74 | 0.458 | −0.2125 | 0.4712 |
| | FAM | $\alpha_4$ | −0.3013 | 0.7398 | 0.2762 | −1.09 | 0.275 | −0.8427 | 0.2400 |
| | Risk | $\alpha_5$ | −0.8078 | 0.5858 | 0.1119 | −7.21 | 0.000 | −1.0273 | −0.5883 |
| | WT | $\alpha_6$ | −0.5168 | 0.5963 | 0.1343 | −3.85 | 0.000 | −0.7801 | −0.2536 |
| | Constant | *con* | 3.1188 | 22.6214 | 0.6727 | 4.64 | 0.000 | 1.8002 | 4.4375 |

## 2.3.2. Phase 1: Allocate the Parking Demand from Origin to the Parking Lot

The parking demand set of origin $o$ is denoted as $PD_o(PA_k, AT_k, PT_k)$, $o = 1, 2, \ldots, O$, and $k = 1, 2, \ldots, K$, where $PA_k$ is the personal attributes set of user $k$, $AT_k$ is the arrival time of user $k$, and $PT_k$ is the parking time of user $k$. This parking demand combines the parking data together. Therefore, we can allocate the user to a parking lot with the personal attributes and allocate it to a parking space with the parking data.

The personal attributes set of the user $k$ is denoted as $PA_k(AGE_k, DE_k, INC_k, FA_k, RISK_{kn}, WT_{kn})$, where $AGE_k$, $DE_k$, $INC_k$, and $FA_k$ represent the age, driving experience, income, and familiarity with

the surrounding parking lot of user $k$, respectively. $RISK_{kn}$ and $WT_{kn}$ mean the perceived risk of no parking space and perceived waiting time for user $k$ to parking lot $n$, respectively.

The parking lot attributes set of parking lot $n$ is denoted as $PLA_n(PF_{nl}, PF_{nm}, TT_{on})$, where $PF_{nl}$ represents the parking fee in the $l^{th}$ open window for parking lot $n$, $PF_{nm}$ is the parking fee in the $m^{th}$ non-window time of parking lot $n$, and $TT_{on}$ means the travel time from the origin $o$ to parking lot $n$.

The utility for user $k$ to travel from origin $o$ to parking lot $n$ at time $t$ is denoted as $U_{on}^{kt}$, and time $t$ is in the $l^{th}$ window time of parking lot $n$.

$$U_{on}^{kt} = \alpha_1 AGE_k + \alpha_2 DE_k + \alpha_3 INC_k + \alpha_4 RISK_{kn} + \alpha_5 WT_{kn} + \beta_1 PF_{nl} + \beta_2 TT_{on} + con \tag{5}$$

Then, the probability for user $k$ to travel from origin $o$ to parking lot $n$ at time $t$ is $P_{on}^{kt}$, as expressed by:

$$P_{on}^{kt} = \frac{e^{U_{on}^{kt}}}{\sum\limits_{n=1}^{N} e^{U_{on}^{kt}}} \tag{6}$$

The parking lot number the user $k$ will be allocated is given by:

$$n = \max(P_{on}^{kt}) \tag{7}$$

### 2.3.3. Phase 2: The Allocation Procedure Based on Parking Lot Rules

Through Phase 1, each user is allocated from the origin to the prior parking lot. Denote the parking demand in parking lot $n$ as $PD_n(PA_k, AT_k, PT_k, M_k)$, then the personal attributes set of the user $k$ in parking lot n is $PA_n(FP_k, AGE_k, DE_k, INC_k, FA_k, RISK_{kn}, WT_{kn})$. Since the parking demand can be obtained through Phase 1, only two variables are added in the $PD_n$ and $PA_n$. $M_k$ represents whether the user is the M-user of the parking lot; if yes, $M_k = 1$, and if not, $M_k = 0$. $FP_k$ is the number of the prior parking lot of user $k$; $FP_k = 1,2,...,N$. This variable is used to record the parking lot number when the user cannot park in the prior parking lot and has to transfer to other parking lots.

Two steps in Phase 2 are implemented to allocate the user in parking to $PRM_n$.

Step 1: Denote the parking demand of parking lot $n$ at time $t$ as $PD_n^t$, and use the PRMM to allocate $PD_n^t$ to the parking space at time $t$; when the user cannot enter the space, go to Step 2.

Step 2: Denote the overflow parking demand as $OD_n^t$, use the MNL model to allocate the user to the next parking lot, then use the PRMM to allocate the user to a space of this parking lot; if the user still cannot find a space, then repeat Steps 1 and 2 until a space is found.

When Steps 1 and 2 are finished, then $t = t + 1$; when $t > T$, then go to the allocation procedure in parking lot $n + 1$, until $n > N$, as shown in Figure 1.

First, input the allocated parking demand in Phase 1, the open window of each parking lot, personal attributes of each user, and parking lot attributes. In Step 1 of Phase 2, when a user is allocated to a parking space, seven different scenarios can be expected:

Scenario 1: Non-window time, parking space is available, and the user is an M-user, then allocate the user to the space in the sequence of space number.

Scenario 2: Non-window time, parking space is available, and the user is a P-user, then the user is not allowed to enter, allocate the user to the next parking lot with the highest probability.

Scenario 3: Non-window time, parking space is unavailable, then no matter whether the user is an M-user or P-user, they are not allowed to enter, allocate the user to the next parking lot with the highest probability.

Scenario 4: Window time, non-reserved space is available, then no matter whether the user is an M-user or P-user, allocate the user to the space in the sequence of space number.

Scenario 5: Window time, non-reserved space is unavailable, reserved space is available, and the user is an M-user, allocate the user to the space in the sequence of space number.

Scenario 6: Window time, non-reserved space is unavailable, reserved space is available, and the user is a P-user, the user is not allowed to enter and is allocated to the next parking lot with the highest probability.

Scenario 7: Window time, both non-reserved and reserved spaces are unavailable, then no matter whether the user is an M-user or P-user, it is not allowed to enter and is allocated to the next parking lot with the highest probability.

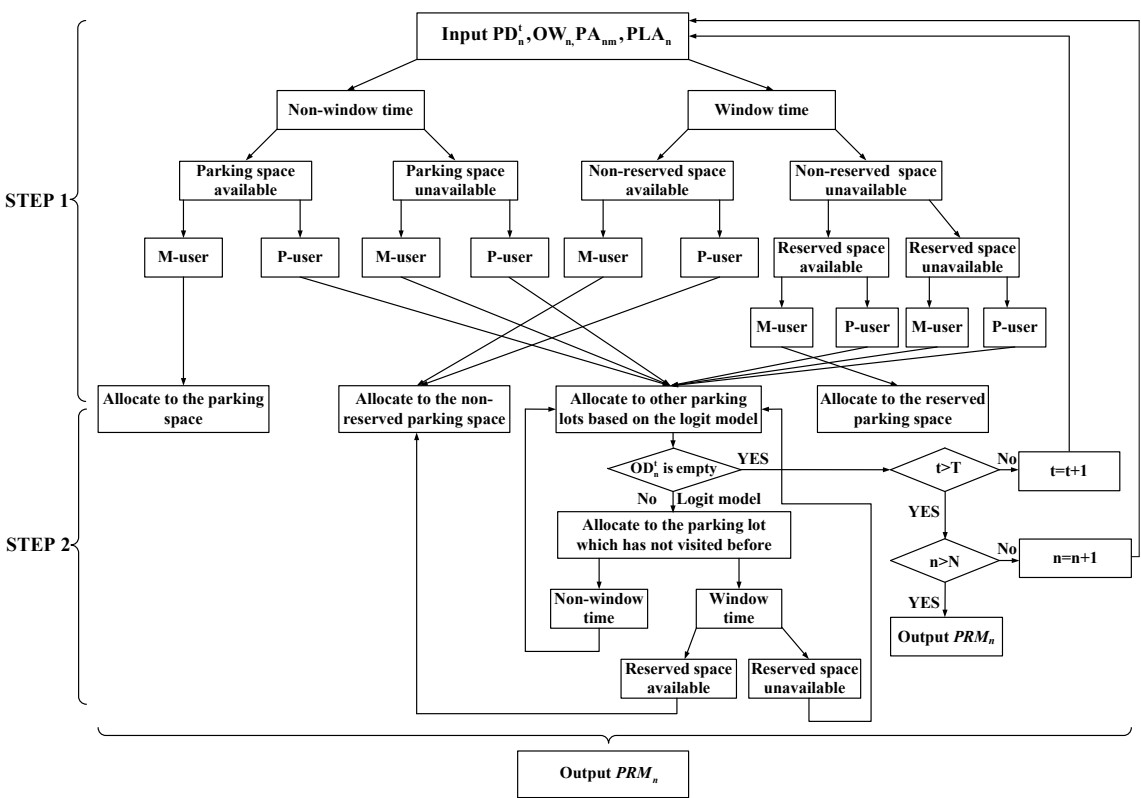

**Figure 1.** The detailed procedure of Phase 2.

We need to distinguish the non-reserved parking space and reserved parking space. Since the number of parking spaces of parking lot $n$ is $p_n$, if the reservation ratio of parking lot $n$ at the $l^{th}$ time window is denoted as $RR_{nl}$, then the non-reserved parking space number ranges from 1 to round$(p_n * (1 - RR_{nl}))$, and the reserved parking space number ranges from round$(p_n * (1 - RR_{nl})) + 1$ to $p_n$.

The parking demand of the user in Scenarios 2, 3, 6, and 7 should be stored in the overflow parking demand set at time $t$ of parking lot $nOD_n^t(PA_k, AT_k, PT_k)$, and go to Step 2.

In Step 2, the overflow user in $OD_n^t$ is allocated to the parking lot with the highest probability and the lot has not been visited before. Since all the users in Step 2 are P-users for other parking lots, there will be three scenarios:

Scenario 1: The arrival time of user $k$ is the non-window time of parking lot $n$, then the user is not allowed to enter; go back to the logit model and allocate the user to other parking lots.

Scenario 2: The arrival time of user $k$ is the window time of parking lot $n$, and non-reserved space is available, then allocate the user to the space in the sequence of space number.

Scenario 3: The arrival time of user $k$ is the window time of parking lot $n$, and non-reserved space is unavailable, then the user is not allowed to enter, go back to the logit model and allocate it to other parking lots.

When all the parking demands in parking lot $n$ at time $t$ are allocated to a parking space, then $t = t + 1$, repeat the allocation procedure until the time reaches $T$. After parking demand of parking

lot *n* is allocated, then $n = n + 1$, until the parking demand of all the parking lots is allocated, end the allocation procedure.

It must be noted that the method proposed in this paper considers the most extreme cases. The method is only applicable to the situation where the surrounding parking lots are all appertaining parking lots of public buildings, the P-users are not allowed to enter at non-window times, and the users do not know the occupancy situation and the window time of the parking lot. However, this method can be easily applied to the general situation by modifying the rules. For example, in the case of public parking lots, the parking lot does not distinguish M-users or P-users, and the open window and reservation spaces are ignored as well. If a user is aware of the occupancy situation of a parking lot, then when allocating the user to a parking lot, the method will determine whether the user can enter the parking lot or not, then allocated to the parking lot with vacant space.

## 3. Algorithm

When all the parking demands are allocated from the origins to a parking space, the information of the allocated $PRM_n$ can help determine whether the open window, parking fee, and reservation ratio of each parking lot are suitable for all the parking lots in the area. To find the optimal solution of each parking lot, an NSGA-II algorithm is proposed based on the PSAM.

The variables in this study include the parking lot attributes set $PLA_n$ ($PF_{nl}$, $PF_{nm}$, $TT_{on}$), the open window $OW_n$, the reservation ratio $RR_{nl}$, the parking demand of each origin $PD_o$ ($PA_k$, $AT_k$, $PT_k$), and personal attributes set $PA_k$. In $PLA_n$, the non-window time parking fee $PF_{nm}$ is the same as the parking fee before implementing shared parking. As the link congestion is not considered in this paper, the travel time $TT_{on}$ between origin and the target parking lot is a constant. $OW_n$ is obtained by the method in Section 2.2. For $PD_o(PA_k, AT_k, PT_k)$, it contains two parts, i.e., the personal attributes and the parking lot attributes. Since there is no precise way to combine one user's personal attributes with their arrival time and parking time together, the personal attributes are selected randomly from the surveyed data of the MNL model and the predicted parking demand is generated randomly with a normal distribution. Then we combine the selected $PA_k$ with the generated ($AT_k$, $PT_k$) together as the parking demand of the origin *o*.

Now, the control variables are only the parking fee of each window time $PF_{nl}$, and the reservation ratio of each window time $RR_{nl}$. $PF_{nl}$ is the most significant variable when a user choosesa parking lot, and $RR_{nl}$ is the critical variable to distinguish the parking space thatanM-user and P-user will be allocated to. In this section, a solution set contains the $PF_{nl}$ and $RR_{nl}$ of each parking lot.

To compare the solution set and determine the optimal parking strategy of each parking lot, the following indices are selected:

1. The rejection number of M-users for parking lot *n* ($f_{1n}$).

This is the most important index that represents how many M-users of parking lot *n* cannot enter the parking lot. Since in $PA_k(FP_k, AGE_k, DE_k, INC_k, FA_k, RISK_{kn}, WT_{kn})$ the $FP_k$ is the prior parking lot number of user *k*, then the $f_{1n}$ is to count the number of users with $FP_k = n$, $M_k = 1$ and allocated to other parking lot at last.

2. The profit of parking lot $n(f_{2n})$.

$$f_{2n} = \sum_{t=1}^{T} (PF_{nm} * (p_n - account0(PRM_n(t,:))) * I_{nw} * I_m + PF_{nl} * (p_n - account0(PRM_n(t,:))) * I_w * I_l) \quad (8)$$

where $I_{nw} = \begin{cases} 1 & t \notin OW_n \\ 0 & t \in OW_n \end{cases}$, $I_w = \begin{cases} 1 & t \in OW_n \\ 0 & t \notin OW_n \end{cases}$, $I_l = \begin{cases} 1 & t \in l\text{th window time} \\ 0 & else \end{cases}$, $I_m = \begin{cases} 1 & t \in m\text{th non} - \text{window time} \\ 0 & else \end{cases}$.

3. The occupancy rate of parking lot $n(f_{3n})$.

$$f_{3n} = \frac{p_n * T - \sum\limits_{t=1}^{T} account0(PRM_n(t,:))}{p_n * T} \tag{9}$$

For parking lot $n$, its parking strategy can be simplified through three indices. The overall evaluation index is denoted as $\Delta_n$. To calculate the overall index, a Z-score normalization method is employed. The $k^{th}$ evaluation index in the $i^{th}$ solution set of parking lot $n$ is denoted as $f_{kn}^i$. Since each parking lot expectsa lower number of rejections, but higher profits and occupancy rate, then the normalized indices are expressed as:

$$f_{kn}^i = \begin{cases} \frac{f_{kn}^i - \overline{f_{kn}^i}}{std(f_{kn}^i)} & k = 1 \\ \frac{\overline{f_{kn}^i} - f_{kn}^i}{std(f_{kn}^i)} & k = 2,3 \end{cases} \tag{10}$$

Suppose the weights of the $k^{th}$ index of parking lot $n$ are $\alpha_{kn}$, then the index of the $i^{th}$ solution for parking lot $n$ is given by:

$$\Delta_n^i = \sum_{k=1}^{3} \alpha_{kn} * f_{kn}^i \tag{11}$$

When implementing shared parking, each parking lot wants to have a lower evaluation value. Therefore, this problem becomes a multi-objective optimization problem with anobject number of $n$, which can be formulated as:

$$\begin{aligned} & min \ (\forall \Delta_n) \\ & n = 1, 2, \ldots, N \\ & s.t. \ \forall PF_{nl} \geq 0 \\ & \quad 0 \leq \forall RR_{nl} \leq 1 \\ & \quad \text{Parking space allocation method} \end{aligned} \tag{12}$$

This problem involves both the multi-objective optimization problem and the parking space allocation. Therefore, a genetic algorithm is a potentially feasible algorithm. For multi-objective optimization problems, the NSGA-II algorithm is the most widely used algorithm, and mainly consists of three parts: (1) a fast non-dominated sorting approach; (2) crowding distance; and (3) Crowded Comparison Operator [27]. To find the optimal solution set of each parking lot, an algorithm which combines the parking space allocation method with NSGA-II is proposed. The whole flow of this algorithm is displayed in Figure 2.

First, input the variables except $PF_{nl}$ and $RR_{nl}$, initialize the first population $POP(1)$, which consists of $N$ individuals, with each individual as a solution set of $PF_{nl}$ and $RR_{nl}$. In the NSGA-II, the main loop starts from the second generation, the new generation $POP(x)$ with size $N$ is selected from the union of $POP(x)$ and $POP(x-1)$ with size $2N$ by fast non-dominated approach, crowding distance calculation and crowded comparison operator. After allocating all the parking demand to $PRM_n$ by taking each individual in $POP(x)$ as the variables, the individuals will go through the procedures of selection, crossover, and mutation, and generate the $POP(x + 1)$. Repeat the procedure of selecting a new $POP(x + 1)$ from the union of $POP(x + 1)$and $POP(x)$ until the number of generation is larger than $X$. Then, the last generation with size $N$ is the final result of the algorithm.

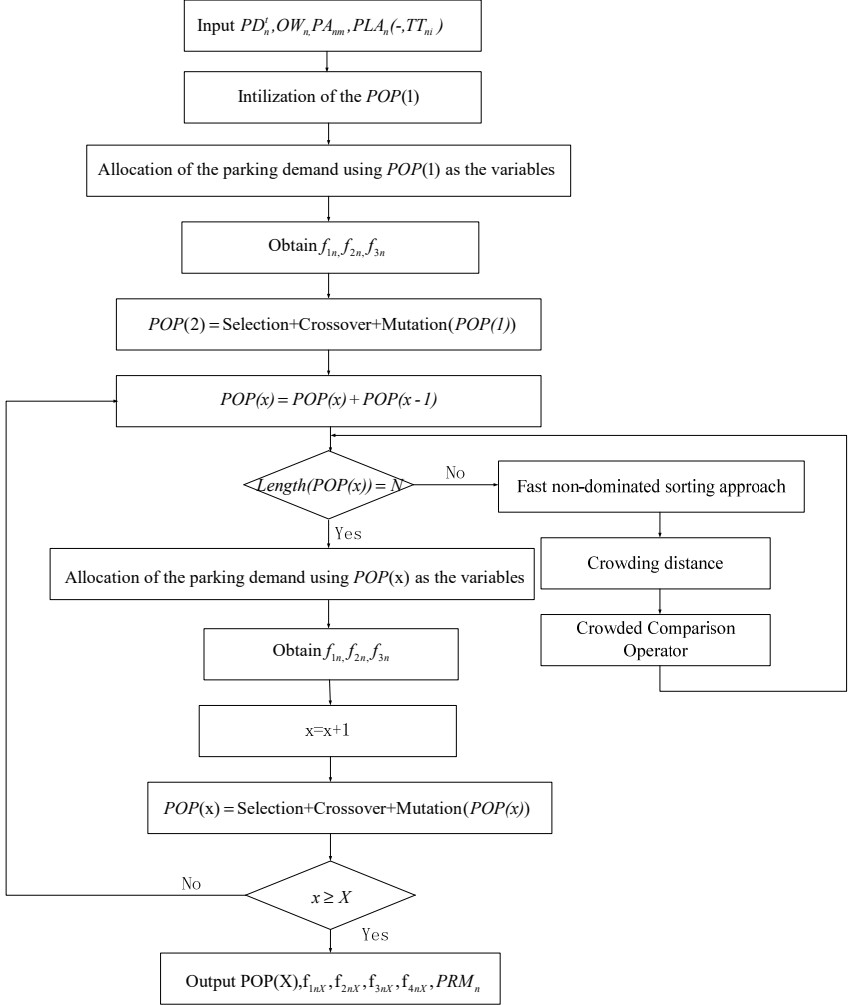

**Figure 2.** The NSGA-II algorithm based on parking space allocation method.

In the last generation, there are *N* solutions in the population, and the optimal solution is among the Pareto front where no solution can dominate other solutions. To find an appropriate solution from the Pareto front, the weight of each parking lot $\beta_n$ is determined by the number of parking spaces and the optimal solution will be chosen with the minimum $\Delta_i^*$ from the Pareto front. $\Delta_i^*$ is given by:

$$\Delta_i^* = \sum_{n=1}^{N} \beta_n \Delta_n^i \ \mathrm{i} = 1, 2, \ldots, \mathrm{I} \tag{13}$$

where *i* is the number of solutions in the Pareto front.

## 4. Experiment

### 4.1. The Gate Data Processing

The appertaining parking lots of five typical public buildings were chosen as the research objects, including hotel, residence, shopping mall, hospital, and government (the residential parking lot is a faculty dorm managed by a high school). The specific sources of these data are listed in Table 3.

**Table 3.** The parking lots of five typical public buildings.

| Type | Survey Parking Lot | Parking Spaces | Survey Time |
|---|---|---|---|
| Hotel | NanjingJinling Star Hotel | 102 | 1 May 2015 to 31 May 2015 |
| Residence | Yangzhong faculty dorm | 98 | 11 Dec 2014 to 27 Dec 2014 |
| Shopping mall | NanjingXinbai Mall | 200 | 1 May 2015 to 17 May 2015 |
| Hospital | Nanjing First Hospital | 179 | 1 May 2015 to 31 May 2015 |
| Government | Nanjing Traffic Authority | 145 | 10 Nov 2014 to 23 Nov 2014 |

The survey was conducted from Thursdays at 12 p.m. to Sunday at 12 p.m., and the time interval of the survey was 1 h. The experiment covered the continuous parking characteristics of the weekday and weekend. Take the hotel parking lot as an example. The allocated $PRM_n$ of the Jinling Star Hotel is illustrated in Figure 3. As shown in the figure, the $a_{ij}$ of different users in the same space are distinguished by 1 and 2; the red line represents the number of parking spaces. As can be seen, from 5 p.m. to 8 p.m. on Saturday the occupancy rate was beyond 100%. This situation also occurred in the other four parking lots. Therefore, it can be deduced that the parking lot can actually accommodate more vehicles than the survey number. However, this overflow phenomenon is not significant and can be ignored in this study.

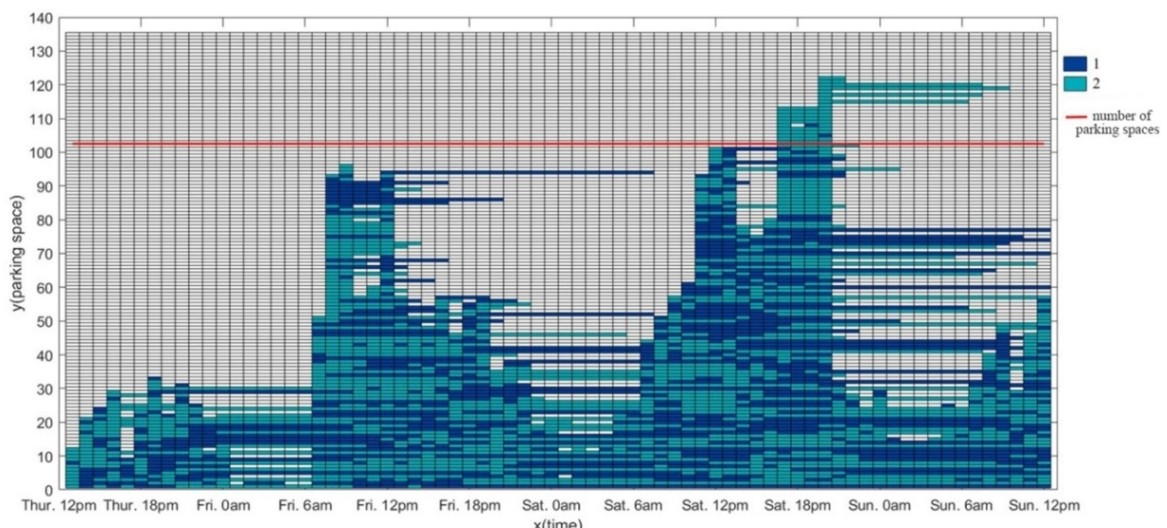

**Figure 3.** The allocation results of the Nanjing Jinling Star Hotel.

The occupancy rate of each parking lot with time is shown in Figure 4.

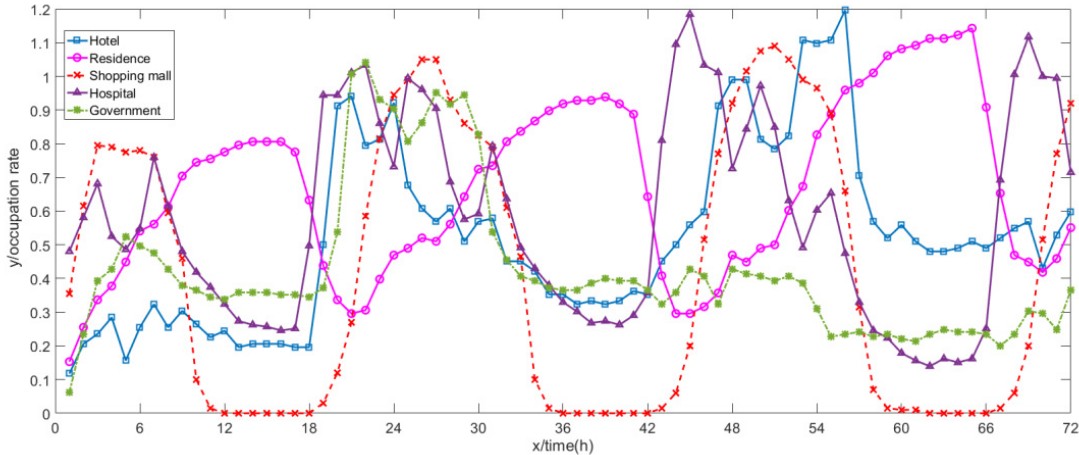

**Figure 4.** Occupancy rate calculated by parking resource matrix method (PRMM).

There is a complementary relationship between the parking lots at peak time and valley time. At night, the peak hour of residential parking lot was the valley time of hotel, shopping mall, and hospital. In the daytime, the valley time of the residential parking lot was the peak hour of other parking lots. Additionally, at weekend, the government parking lot was in the valley period for the whole day, and a large number of shared parking spaces were available for users.

An important assumption when using PRMM to allocate the gate data to $PRM_n$ is that each user will be allocated to the space in the sequence of parking space number, while in the real world, users will not do that. In the traditional method, the occupancy rate of a parking lot at time $t$ is calculated as the current number of users plus the number of arrived users minus leaving users. The occupancy rate calculated by the traditional method is illustrated in Figure 5.

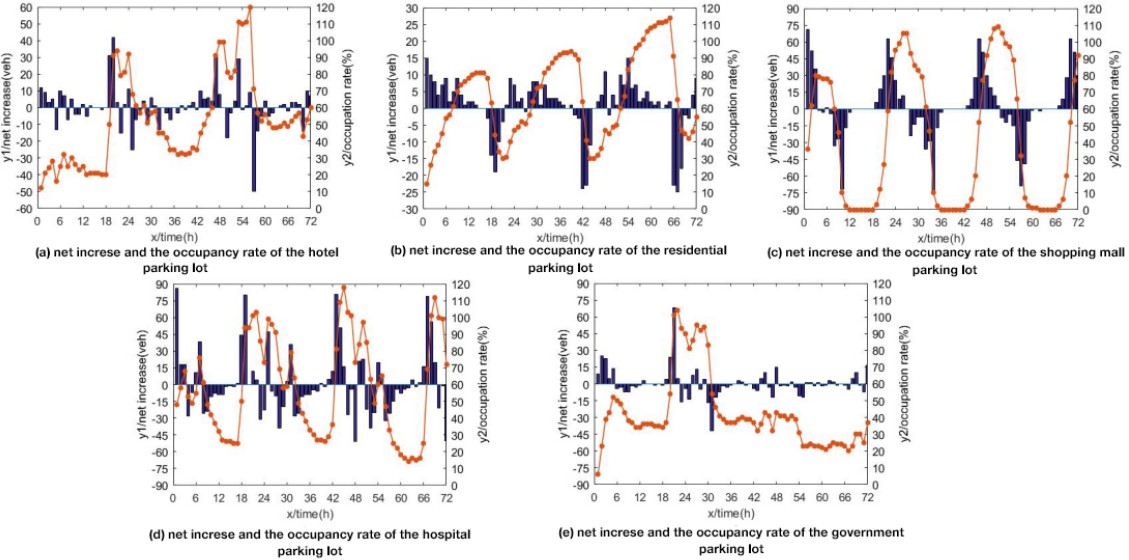

**Figure 5.** The occupancy rate calculated by the traditional method.

The left ordinate in Figure 5 is the net increase in the number of the users at time $t$, which is obtained by the arrived number minus the leaving number; the right ordinate is the occupancy rate. It can be clearly seem from the figure that the occupancy rate obtained using PRMM is exactly the same as that obtained using the traditional method, which verifies the accuracy of PRMM.

After obtaining the allocated PRM, suppose the minimum duration of open window is 6 h, the minimum number of spaces that should be provided for P-users is expressed as $m_1 = 0.3 \times p_n$. The open window of each parking lot is shown in Table 4.

**Table 4.** The open window of each parking lot.

| Type | Open Window |
|---|---|
| Hotel | Thur.12:00~Fri.7:00, Fri.13:00~Sat.10:00, Sat.22:00~Sun.12:00. |
| Residence | Thur.12:00~Thur.20:00, Fri.6:00~Fri.17:00, Sat.6:00~Sat.17:00. |
| Shopping mall | Thur.21:00~Fri.10:00, Fri.20:00~Sat.11:00, Sat.21:00~Sun.10:00. |
| Hospital | Thur.21:00~Fri.6:00, Fri.20:00~Sat.6:00, Sat.20:00~Sun.6:00 |
| Government | Thur.12:00~Fri.8:00, Fri.18:00~Sun.12:00 |

*4.2. Calculation Results of the Parking Space Allocation Method*

Suppose there are five parking lots in one area, which are of the hotel, residence, shopping mall, hospital, and government, which are numbered by 1–5, respectively. Each parking lot does not allow P-users to enter at non-window times. There are five dummy origins around the area. The network containing the five parking lots and dummy origins are shown in Figure 6.

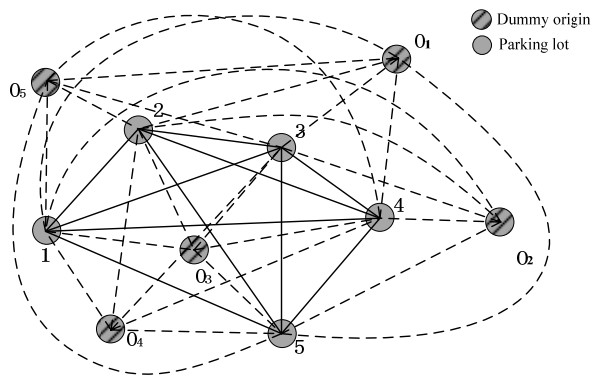

**Figure 6.** The road network of the experiment.

The road network between each parking lot is simplified to one link, and each dummy origin is directly connected to all the parking lots. In this case, the travel time of each link is shown in Table 5.

**Table 5.** The travel time from dummy origin to parking lots and travel time between parking lots (min).

|  | $P_1$ | $P_2$ | $P_3$ | $P_4$ | $P_5$ |  | $P_1$ | $P_2$ | $P_3$ | $P_4$ | $P_5$ |
|---|---|---|---|---|---|---|---|---|---|---|---|
| $O_1$ | 13 | 8 | 6 | 7 | 10 | $P_1$ | - | 4 | 6 | 9 | 4 |
| $O_2$ | 15 | 11 | 7 | 4 | 6 | $P_2$ | 4 | - | 3 | 5 | 8 |
| $O_3$ | 4 | 5 | 5 | 6 | 4 | $P_3$ | 6 | 3 | - | 4 | 6 |
| $O_4$ | 3 | 6 | 8 | 9 | 4 | $P_4$ | 9 | 5 | 4 | - | 4 |
| $O_5$ | 4 | 3 | 7 | 11 | 9 | $P_5$ | 4 | 8 | 6 | 4 | - |

The parking fee for each parking lot at non-window times is presented in Table 6.

**Table 6.** The parking fee of each parking lot at non-window time (RMB).

| Parking Lot | Open Window and Parking Fee | | |
|---|---|---|---|
| Hotel | Fri.8:00~Fri.12:00 | Sat.11:00~Sat.18:00 | Sat.19:00~Sat.21:00 |
| | 10 for the first hour, then 2 per hour | | 5 for the first hour, then 1 per hour |
| Residence | Thur.21:00~Fri.5:00 | Fri.18:00~Sat.5:00 | Sat.18:00~Sun.12:00 |
| | 3 for the first hour, then 1 per hour | | |
| Shopping mall | Thur.12:00~Thur.20:00    Fri.11:00~Fri.19:00    Sat.12:00~Sat.20:00 | | Sun.11:00~Sun.12:00 |
| | 2 per hour | | |
| Hospital | Thur.12:00~Thur.21:00    Fri.7:00~Fri.19:00    Sat.7:00~Sat.19:00 | | Sun.7:00~Sun.12:00 |
| | 10 for the first hour, then 1.5 per half hour | | |
| Government | Fri.9:00~Fri.17:00 | | |
| | Free for M-users | | |

The gate data used to determine the open window is the data before implementing shared parking, and the demand from M-users will not change too much after implementing shared parking. As a result, the gate data of the second week was used as one part of the parking demand of each parking lot, and the parking demand of each dummy origin was generated randomly with the uniform distribution.

Since there are 14 window times for all the five parking lots, each open window has a parking fee and a reservation ratio, so the total number of variables is 28. The population size was set as 100, the cross probability as 0.8, and the mutation probability as 0.05. The weights of three indices, i.e., rejection number, parking profits, and occupancy rate were set as 0.5, 0.25, and 0.25, respectively. The allocation process was run for 3000 generations [28]. In the last generation, the Pareto front contained 56 solutions. Table 7 displays the solution with the minimum $\Delta^*$ and its evaluation indices.

**Table 7.** The results of the optimal parking strategy for each parking lot.

| Reservation Ratio | | | Parking Fee | | Rejection Number (f$_{1n}$) Profit (f$_{2n}$) Occupancy Rate (f$_{3n}$) | | | Weighted Index Value | |
|---|---|---|---|---|---|---|---|---|---|
| Hotel | $RR_{11}$ | 0.13 | $PF_{11}$ | 2 | Hotel | $f_{11}$ | 0 | $\Delta_1$ | 0.100208 |
| | $RR_{12}$ | 0.18 | $PF_{12}$ | 1 | | $f_{21}$ | 10489 | $\Delta_2$ | 0.201779 |
| | $RR_{13}$ | 0.11 | $PF_{13}$ | 2 | | $f_{31}$ | 0.8380 | $\Delta_3$ | 0.309357 |
| Residence | $RR_{21}$ | 0.26 | $PF_{21}$ | 3 | Residence | $f_{12}$ | 25 | $\Delta_4$ | 0.233041 |
| | $RR_{22}$ | 0.28 | $PF_{22}$ | 3 | | $f_{22}$ | 13209 | $\Delta_5$ | 0.155616 |
| | $RR_{23}$ | 0.33 | $PF_{23}$ | 4 | | $f_{32}$ | 0.9040 | $\Delta^*$ | 0.215671 |
| Shopping mall | $RR_{31}$ | 0.10 | $PF_{31}$ | 1 | Shopping mall | $f_{13}$ | 34 | | |
| | $RR_{32}$ | 0.15 | $PF_{32}$ | 1 | | $f_{23}$ | 9746 | | |
| | $RR_{33}$ | 0.11 | $PF_{33}$ | 1 | | $f_{33}$ | 0.6768 | | |
| Hospital | $RR_{41}$ | 0.28 | $PF_{41}$ | 2 | Hospital | $f_{14}$ | 28 | | |
| | $RR_{42}$ | 0.24 | $PF_{42}$ | 2 | | $f_{24}$ | 27051 | | |
| | $RR_{43}$ | 0.31 | $PF_{43}$ | 3 | | $f_{34}$ | 0.7595 | | |
| Government | $RR_{51}$ | 0.11 | $PF_{51}$ | 2 | Government | $f_{15}$ | 15 | | |
| | $RR_{52}$ | 0.05 | $PF_{52}$ | 3 | | $f_{25}$ | 21007 | | |
| | | | | | | $f_{35}$ | $f_{35}$ | | |

From the solution for the hotel parking lot, it can be seen that, since the parking fee was at a high level in non-window times, to attract more users, the parking fees at window times were level 2,1, and 2, respectively. For the residential parking lot, since the window time was concentrated in the daytime, and this time period was the peak time of other parking lots, the highest occupancy rate of 92% was obtained. To protect the M-users of residential parking lot, the reservation ratio was between 0.26~0.33 while the parking fee was at the highest level. For the shopping mall parking lot, since in the early morning there was barely any parking demand from M-users, the reservation ratio and parking fee were very low. For the hospital parking lot, the parking fee before implementing shared parking was very high, which made the profits of the hospital the highest; at open window times, the parking fee was changed to a normal level; though it might lead to a decline in profits, the occupancy rate increased at the same time. For the government parking lot, since it could provide a large number of spaces throughout the whole weekend, the reservation ratio at the second window time was low; even when the parking fee increased to a high level, there were still a large number of users going to park in the government parking lot.

The distribution of different users, the rejection number of M-users, profits, and occupancy rate are illustrated in Figure 7, Figure 8, Figure 9, and Figure 10, respectively.

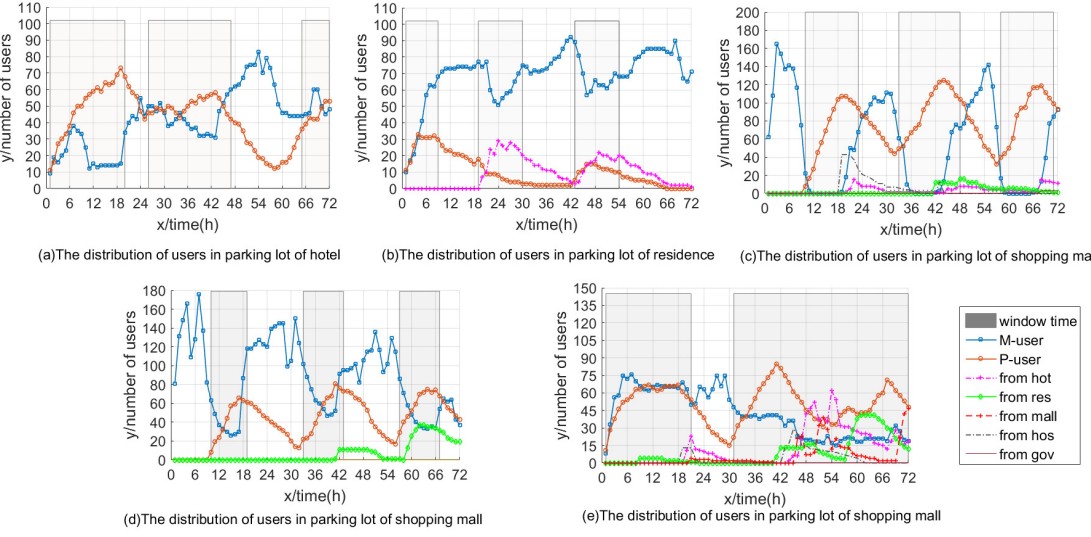

**Figure 7.** The distribution of users in each parking lot.

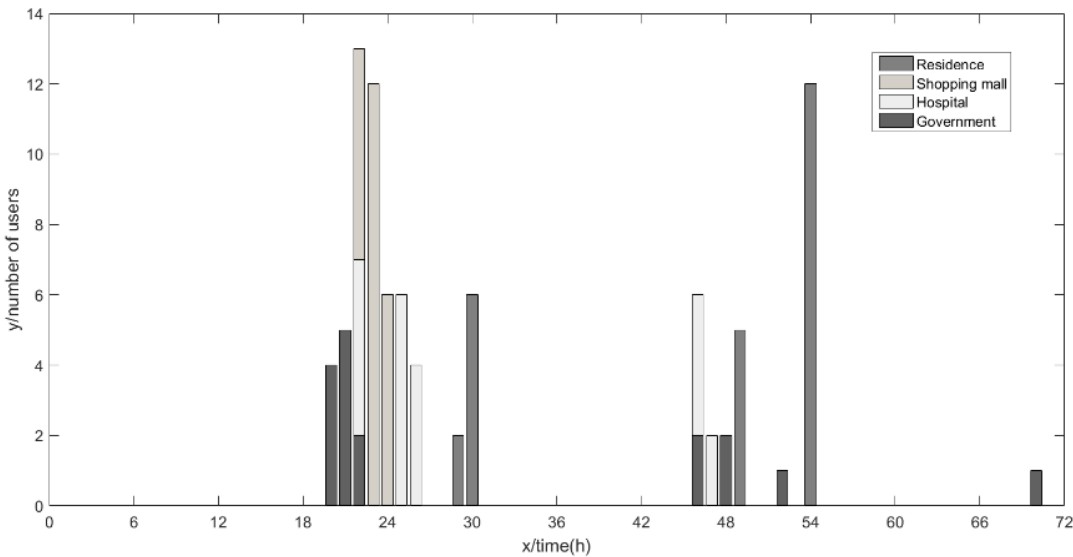

**Figure 8.** The distribution of the rejected M-users.

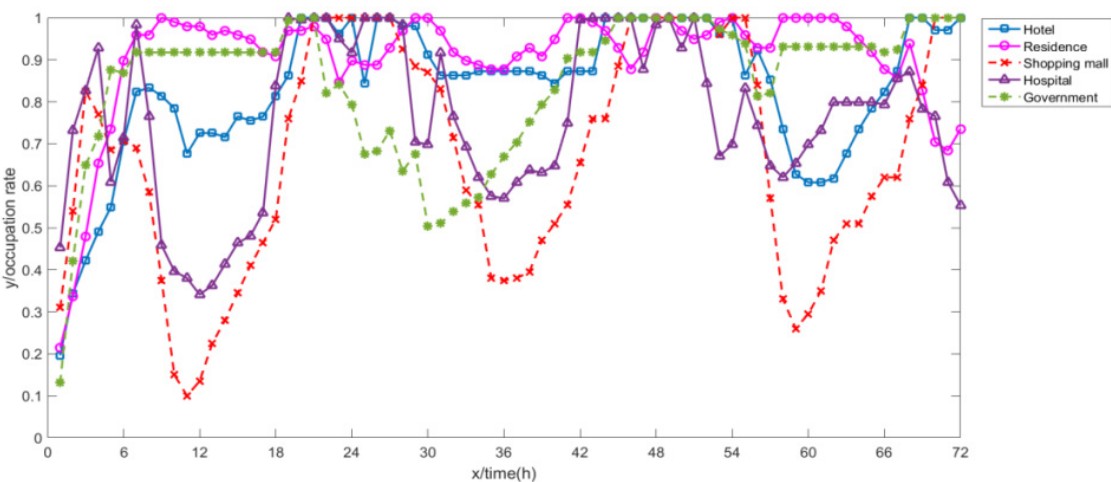

**Figure 9.** The occupancy rate of each parking lot.

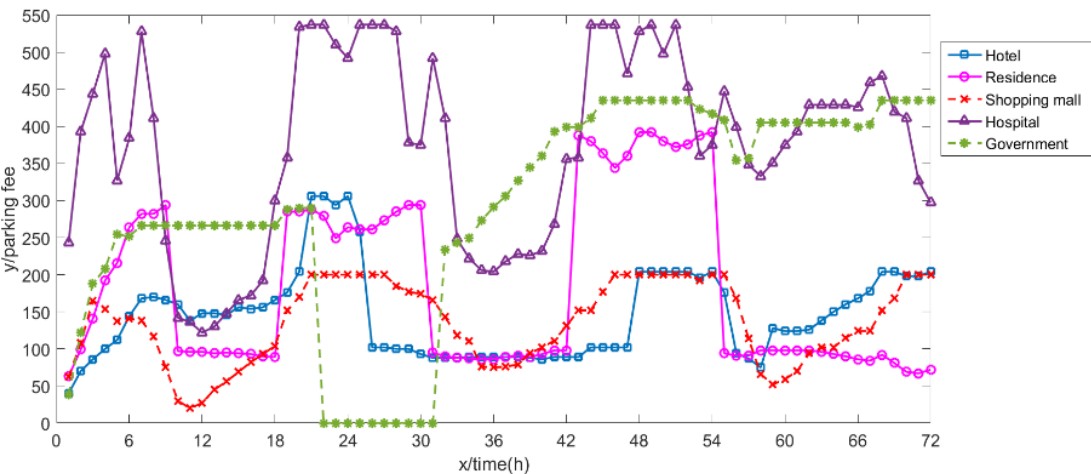

**Figure 10.** The profit of each parking lot.

As can be seen from Figure 7, from hot means the prior parking lot was the hotel's, however the user could not enter the hotel parking lot, and finally found a space in other parking lots. The results for from res, mall mall, hos, and gov have the same meaning. In Figure 7, since most of the data

of M-users were from the gate data of the second week, there was no big difference between the distribution of the first-week data and second-week data. It can be found that the distribution of M-users was similar to the occupancy rate of the first week in Figure 4, where only 0, 25, 34, 28, and 15 M-users of the five parking lots were rejected in three days, which was an acceptable number.

In Figure 7a, at the first window time, from 1 to 20h, the number of P-users increased. From the end of the first window to the start of the second window, the number of P-users decreased. Since the non-window time between the two window time is too short, although the P-users are not allowed to enter at non-window times, there were still some P-users who had not left the parking lot. This situation was also observed in the other four parking lots.

In Figure 7b, the reservation ratio was high in the residence parking lot, making the number of P-users lower than that of other four parking lots. In comparison, in Figure 7e, the reservation ratio was low in the second window time of the government parking lot, which led to a high occupancy rate of P-users and users from other parking lots.

In Figure 8, it can be seen that the times during which M-users could not enter were concentrated at 1 or 2 h after the end of window time, during which period some P-users had not left and occupied the parking spaces of M-users. However, in other time periods, most of the M-users were able to find parking spaces.

In Figure 9, it can be seen that most of the time, the occupancy rate was relatively balanced in the five parking lots. Three time periods from late night to the early morning of the next day—8–18h, 30–44h, and 55–66h—were the valley times of the hotel, shopping mall, and hospital, respectively. When the total demand was limited in these three time periods, the occupancy rate was not high.

As can be seen from Figure 10, the profits of each parking lot were related to the parking fee, occupancy rate, and the number of parking spaces. For the hospital parking lot, since the parking fee was always at a relatively high level and the number of parking spaces was 179 (the second highest of the five parking lots) the profits of the parking lot were the highest. In comparison, although the shopping mall had the most parking spaces (200), the occupancy rate was too low at night, and the parking fee was at the lowest level, leading to this parking lot making the lowest profit of the five parking lots.

### 4.3. Comparison with All Shared Parking Strategy

In order to further evaluate the accuracy of the calculation results, a comparison was performed between the proposed method and the traditional "all time all space" shared parking strategy. In the traditional shared parking strategy, all time periods were window times, and the reservation ratio was 0. Since the $\Delta_n$ was normalized according to other solutions in the population, it does not make sense to compare the results obtained by different methods. Only the specific indices of $f_{kn}$ are provided to show which strategy is better, as shown in Table 8.

The results of the PSAM are a non-dominated solution of all shared parking strategy. However, it is clear that there is an imbalance in the results of the traditional strategy. Since the users were allowed to enter at any time in any space, the occupancy rates of the shopping mall, hospital, and government parking lots were very high. It was natural that too many P-users occupied the parking spaces, making the rejection number of M-users to increase to 187, 2396, and 519 for the three parking lots, respectively, which is unacceptable for the parking lot managers.

In all shared parking strategies, the distribution of different users is shown in Figure 11.

In comparison with Figure 7, there was no significant difference in the occupancy situation of M-users in the parking lots of the hotel, residence, and shopping mall. However, the number of P-users and users from other parking lots increased significantly. In Figure 11d, it is obvious that at 18–30h, which was the peak time of the hospital, the number of M-users was very low. The parking spaces for patients were occupied by too many public users, although in other parking lots, the all time shared parking might perform well, except in one area; if one parking lot was filled up with vehicles and influence the normal operation of the building, this strategy cannot be considered as a good one.

**Table 8.** The results of the parking space allocation method (PSAM) and all shared parking strategy.

|  |  | **PSAM** | **All Shared Parking Strategy** |
|---|---|---|---|
| Hotel | $f_{11}$ | 0 | 0 |
|  | $f_{21}$ | 10,489 | 9904 |
|  | $f_{31}$ | 0.8380 | 0.9569 |
| Residence | $f_{12}$ | 25 | 0 |
|  | $f_{22}$ | 13,209 | 9418 |
|  | $f_{32}$ | 0.9040 | 0.9727 |
| Shopping mall | $f_{13}$ | 34 | 187 |
|  | $f_{23}$ | 9746 | 10,028 |
|  | $f_{33}$ | 0.6768 | 0.9172 |
| Hospital | $f_{14}$ | 28 | 2396 |
|  | $f_{24}$ | 27,051 | 21,458 |
|  | $f_{34}$ | 0.7595 | 0.9760 |
| Government | $f_{15}$ | 15 | 519 |
|  | $f_{25}$ | 21,007 | 20,828 |
|  | $f_{35}$ | 0.8478 | 0.9712 |

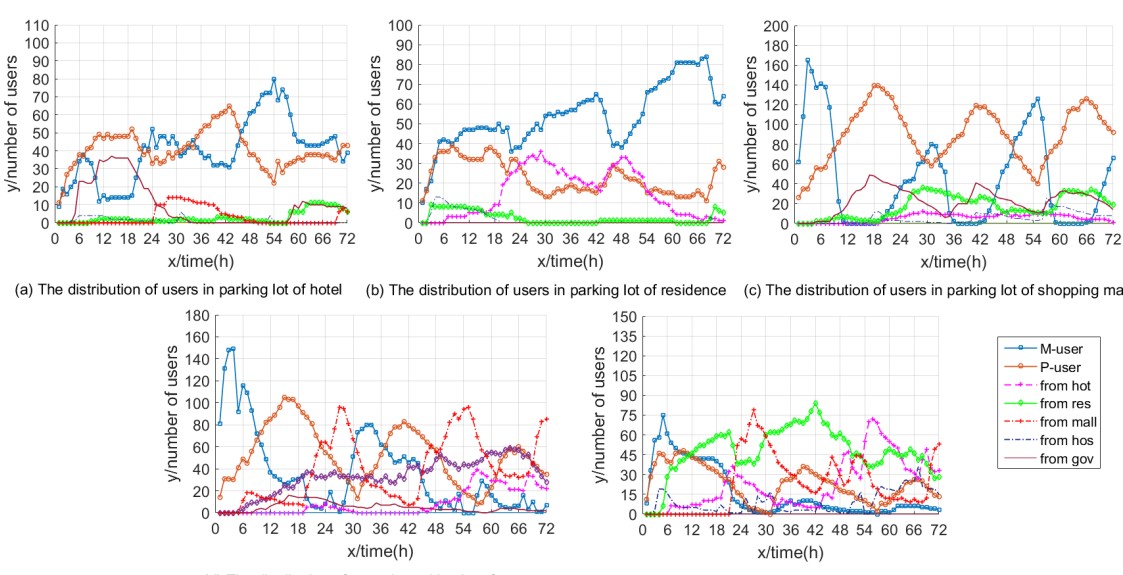

**Figure 11.** The distribution of users in each parking lot calculated by all shared parking strategies.

## 5. Conclusions

In order to make full use of the appertaining parking lots of public buildings, this paper establishes a method to determine the optimal shared parking strategy of each parking lot in the area from the perspective of parking lots and users. For the perspective of parking lots, the open window, parking fee, and reservation ratio were used to ensure the interests of M-users and the parking lots. From the perspective of users, a parking space is chosen based on the user's parking demand and the parking lot's rules. The performance of the proposed method is verified by comparing it with the traditional shared parking strategy.

This study makes the following three significant contributions:

1.　PRMM is proposed to process the gate data, with the function of allocating one user into a specific parking space and obtaining the distribution of users in the parking lot. This method provides a way to integrate parking spaces with other transportation elements such as link, path, and network equilibrium.

2.　The shared parking strategy is considered for all the parking lots in an area to motivate each appertaining parking lot to carry on the shared parking.

3.　A method has been established to allocate the parking demands from origins to the final parking space, where the personal attributes and its parking attributes are both considered.

However, there are also some areas to be improved in this study, such as:

1.　After the end of the open window time, the problem of remaining P-users should be further investigated; during this time period the interest of M-users cannot be guaranteed.

2.　A prediction model is necessary to forecast the attracted demands after implementing shared parking.

Future research will focus on the practical application of this method. A comprehensive investigation of public parking lots, appertaining parking lots, off-street parking, and on-street parking in a certain area should be conducted first. Then, to precisely predict the parking demand after implementing shared parking strategy, since the parking demand is a discrete variable, a prediction model based on negative binomial will be considered in future work [29,30]. After finishing these two work areas, we hope to employ this method to design a specific parking strategy for each parking lot in this area.

**Author Contributions:** Conceptualization, J.C.; Methodology, Y.C.; Software, Y.C.; Validation, Y.C. and B.W.; Formal Analysis, C.Z.; Data Curation, B.W.; Writing-Original Draft Preparation, Y.C.; Writing-Review & Editing, C.Z.; Visualization, Y.C.; Supervision, J.C.; Project Administration, J.C.; Funding Acquisition, J.C.

**Funding:** This research was financially supported by the Key Project of the National Natural Science Foundation of China (Grant No. 51478111 & 51638004).

**Acknowledgments:** We thank the anonymous reviewers for their careful work and insightful comments.

**Conflicts of Interest:** The authors declare no potential conflict of interests.

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
