# Peer review of "A Parking Space Allocation Method to Make a Shared Parking Strategy for Appertaining Parking Lots of Public Buildings"

_sustainability, doi:10.3390/su11010120_

Round 1
Reviewer 1 Report
This is an interesting and well-written paper. Here are some suggestions to improve the paper:
1) In the paper, the full spelling of “MNL” model, “PRM” should be provided when first appear.
2) Page 3, line 99. “This is called parking space allocation method in this paper. (PSAM)” should be “This is called parking space allocation method (PSAM) in this paper.”
3) Algorithm 1 should be further explained.
4) Sub-section title 2.3.1, the logit model usually refers to the logistic model. Thus, the title should be “The multinomial logistic regression model”
5) In the experiment, does the acceptance rate of drivers considered in the analysis? That is, drivers may not follow the parking space allocation assignment.
6) Figure 11, the legend is a little small to observe.
7) For future study of prediction models, since the parking demand is a discrete variable (count data). The negative binomial may be considered as one potential prediction model. Some studies on this topic should be reviewed and discussed as future work. For example: “Negative binomial additive models for short-term traffic flow forecasting in urban areas. IEEE Transactions on Intelligent Transportation Systems, 2014”. “Empirical Bayes estimates of finite mixture of negative binomial regression models and its application to highway safety. Journal of Applied Statistics, 2018, 45(9).”
8) Page 10, line 313, “which is consisted of” should be “which consists of”
9) Page 3, line 92, “In section 4” should be “In Section 4”
Overall, the paper is well written.
Reviewer 2 Report
It is an interesting paper but I am not sure this kind of article fits the purpose of this journal and I rather leave this decision to the editor. It seems to me that this is a rather technical article, maybe meant for a different audience/journal.
First, I would like to state that I lack the statistical knowledge to judge the model in your paper; hopefully, another reviewer will give you some feedback on that.
As a parking expert I can provide you with some feedback over the added value of this paper for the sector:
1) I miss the real research question behind this research; what you refer to as 'appertaining parking' is mostly known in the sector as 'double use of parking facilities'. Usually, the main issues concerned to that have a legal, administrative or operational aspect: is anyone allowed to enter a specific building? is the parking management system equipped to differentiate drivers? what is the administrative burden to arrange double parking for the owner of the building? etc... These are typical questions that we are facing in the parking sector. It seems to me that you are 5 steps ahead of the sector, that's why I am wondering if this is the right journal where to publish. Honestly, I think this paper is too technical and, accordingly, might fit better another journal. The main question you should answer is: who is going to benefit from this paper? I don't see many people working in the parking sector could benefit from it;
2) Please, the paper needs an extensive English editing;
Minor issues:
- in your MNL model you consider personal attributes; why would these attributes have any influence in the choice of allocating parking spots? Especially age and driving experience; you should explain better why you want to use these variables
Round 2
Reviewer 2 Report
I didn't know it was for a special issue on MaaS; I still think it's rather technical for the typical reader of this journal, anyway I leave the decision to the editor.